# Piezoelectric Nanogenerator Based on Electrospinning PVDF/Cellulose Acetate Composite Membranes for Energy Harvesting

**DOI:** 10.3390/ma15197026

**Published:** 2022-10-10

**Authors:** Yuanyuan Li, Qing Hu, Rui Zhang, Wenmei Ma, Siwei Pan, Yaohong Zhao, Qing Wang, Pengfei Fang

**Affiliations:** 1School of Physics and Technology, Wuhan University, Wuhan 430072, China; 2Electric Power Research Institute of Guangdong Power Grid Co., Ltd., Guangzhou 510080, China

**Keywords:** piezoelectric, nanogenerator, electrospinning, polyvinylidene fluoride, cellulose acetate

## Abstract

The organic piezoelectric polymer polyvinylidene fluoride (PVDF) has attracted extensive research because of its excellent flexibility and mechanical energy-harvesting properties. Here, the electrospinning technique was taken to fabricate synthesized fiber membranes of a PVDF/cellulose acetate (CA) composite. The obtained PVDF/CA electrospun fiber membranes (EFMs) were employed to prepare a flexible nanogenerator. XRD and FTIR spectroscopy revealed the enhancement of piezoelectric behavior due to an increase in β-phase in PVDF/CA EFMs compared with cast films. The PVDF/CA fibers (mass ratio of PVDF to CA = 9:1) showed an output voltage of 7.5 V and a short-circuit current of 2.1 μA under mechanical stress of 2 N and frequency of 1 Hz, which were 2.5 and two times greater than those of the pure PVDF fibers, respectively. By charging a 4.7 µF capacitor for 15 min with the voltage generated by the PVDF/CA EFMs, nine LED lamps could be lit. The work provides an effective approach to enhancing the piezoelectric effects of PVDF for low-power electronic loading of macromolecule polymers.

## 1. Introduction

The piezoelectric properties of materials can be used to convert mechanical energy into electricity. Since 2006, when the piezoelectric nanogenerator (PENG) was first introduced by Wang [1], countless nanostructured materials have been designed as PENGs to harvest all kinds of energy from the surrounding environment. The upsurge of related piezoelectric materials consisting of inorganic piezoelectric materials, such as lead zirconate titanate (PZT) [2,3], zinc oxide (ZnO) [4,5], and barium titanate (BaTiO_3_) [6,7,8], has been a focus due to their high-performance piezoelectric properties. However, they are inherently rigid and may contain toxic elements that impede their implantable or wearable applications, respectively. On the other hand, energy harvesters based on organic materials such as poly(vinylidene fluoride) (PVDF) and its copolymers [9,10] have been proposed. The flexible PVDF exhibits excellent piezoelectric, pyroelectric, ferroelectric, and dielectric properties, and its copolymers are considered suitable materials for smart stretchable and wearable electronics. Recent developments in the fabrication of one-dimensional ferroelectric fibers using electrospinning have opened up exciting new opportunities for the design and development of fiber-based actuators and sensors.

PVDF, a representative piezoelectric synthetic polymer, is the most widely used piezoelectric polymer due to its high piezoelectric constant [11,12]. It has previously been observed that PVDF has at least five crystalline phases (α, β, γ, δ, and ε) [13]. Among them, rather than the nonpolar α-phase (TGTG’ conformation), the β-phase (TTTT conformation) has attracted wide attention due to its high polarity and the largest electric dipole moment [14,15], which play a vital role in the piezoelectricity of PVDF. Therefore, various techniques have been developed to drive a higher proportion of the β-phase in PVDF. Extensive research has shown that polarization under a high electrical field in electrospinning can effectively induce the phase change from the nonpolar α-phase to polar β-phase [16,17,18,19]. However, PVDF fibers obtained via electrostatic spinning struggle to generate enough piezoelectric response to meet the actual requirements [20,21], after which they partially depolarize.

Recently, the introduction of organic or inorganic fillers in PVDF has been an effective means to achieve the β-phase, as well as prepare multicomponent composites, such as functionalized multiwalled carbon nanotubes [22,23], nanoclay [24], cellulose nanocrystals [25] and its derivatives, and other macromolecule polymers. The abovementioned methods are aimed at the reorientation of β-phase nanocrystals for the piezoelectricity of PVDF [26].

The high crystallinity and abundant polar hydroxyl groups result in cellulose containing a large number of dipoles with strong electron-donating ability, granting it excellent piezoelectric and friction electrical effects [27,28]. Cellulose acetate (CA) is a derivative of cellulose that reaches a certain degree of acetylation, which has good fiber-forming performance [29]. Because of its nontoxicity, solubility in organic solvents, good stability, biocompatibility, and biodegradability, CA has become a branch of great development potential in electrospinning fiber membranes [30,31,32]. Liu et al. reported PVDF/graphene oxide (GO) nanofibers obtained via an electrospinning process. Chemical interactions such as hydrogen bonds are formed between the CF_2_ groups of PVDF and the −COOH/−OH bonds in the GO lamellae, leading to an all-trans TTTT conformation [33]. In the present case, the addition of CA seems to play a similar role.

In this paper, we prepared electrospun PVDF/CA fiber membranes (EFMs). CA was compounded into PVDF to improve the β-phase content and piezoelectric response, and the morphology and physicochemical properties following the electrospinning process of the composite membranes with different CA contents were discussed in detail. A PVDF/CA-based flexible nanogenerator was also manufactured for harvesting energy.

## 2. Materials and Methods

### 2.1. Materials

N,N-Dimethylformamide (analytical grade) and acetone (analytical grade) were bought from Sinopharm Chemical Reagent Co., Ltd. (Shanghai, China). Polyvinylidene difluoride (Mw 600000, PVDF, New Materials Co., Ltd., Shanghai, China) was bought from 3F New Materials Co., Ltd. (Shanghai, China). Cellulose acetate (CA) (7.3Y36000, Zhuhai Acetate Co., Ltd. Zhuhai, China) was produced by Zhuhai Acetate Co., Ltd. (Zhuhai, China).

### 2.2. Preparation

Firstly, the solvent consisted of a mixture of N,N-dimethylformamide (DMF) solution and acetone solution in a mass ratio of 6:4 PVDF/CA with a total mass fraction of 12 wt.%. Then, different mass fractions of CA and PVDF (mass ratios of 10:0, 9:1, 8:2, and 7:3, recorded as PVDF/CA-1, PVDF/CA-2, and PVDF/CA-3 respectively) were dissolved in the solvent and stirred for 7 h to obtain the PVDF/CA mixed spinning solution. The electrospinning process was performed with a supply voltage of 18 kV and a rotational speed of 1300 r/min. The syringe injection rate was 0.4 mL/h, the distance between the needle and the collector was 15 cm, and the spinning time was 12.5 h. The PVDF/CA fiber membrane was collected on a roller wrapped in aluminum foil at 25 °C and relative humidity of 30% ± 5%. Lastly, the prepared membrane was removed from the roller and put into a drying oven at 60 °C for 8 h.

In the fabrication of cast film, the PVDF/CA mixed solution was prepared using the same method described above. Then, the solution was poured evenly onto an 8 cm × 8 cm dry glass sheet and naturally leveled. The thickness of the cast film was controlled using a squeegee to match the thickness of the prepared EFMs. Lastly, the films were dried in an oven at 60 °C for 10 min.

Using the electrospinning technique, the PVDF/CA fiber membranes were applied to construct PENGs with a typical sandwiched structure. To fabricate the PENGs, two layers of copper foil were applied to the membrane surface as the electrode material. Then, they were encapsulated with PET film to avoid friction across the fibers, electrodes, and encapsulation material. The assembled sandwich-type structure device is shown in Figure 1b.

### 2.3. Measurement of PENGs

A self-made laboratory pressure device driven by a reciprocating motor was applied to the experiments for PENGs (Figure 1a). The voltage outputs were recorded using a digital oscilloscope (DS2202A, Rigol technologies, Suzhou, China), and the short-circuit currents were measured using an electrochemical workstation (CS310, Corrtest instruments, Wuhan, China).

### 2.4. Characterization 

A field-emission scanning electron microscope (SEM, Hitachi, Tokyo, Japan) was used to characterize the morphology of the films. X-ray diffractometry (XRD, Rigaku, Tokyo, Japan) and Fourier-transform infrared spectroscopy (FTIR, Thermo Fisher Nicolet, MA, USA) were used for phase and composition analysis.

## 3. Results and Discussion

Figure 2 shows the SEM images of PVDF/CA fibers with different mass ratios. The fibers were uniform and neat with a smooth surface, and the addition of CA increased the fiber diameter, increased the spatial distribution, reduced the density, and improved the uniformity. As the addition of CA increased the viscosity of the spinning solution, and the solution containing more CA overcame higher internal friction of the liquid to flow, a greater proportion of CA hindered the complete stretching of the spinning solution with a constant electric field force and Coulomb repulsion on the charged jet. Therefore, only CA contents less than 30% relative to the weight of the solute could be used to prepare the PVDF/CA composite fiber membranes via electrostatic spinning, to prevent clogging of the needles and the inability to form fibers.

Using ImageJ software, the diameter distribution of PVDF/CA composite fibers was calculated, as shown in Figure 3. It can be seen that the diameter of the pure PVDF fiber was 0.24 μm, whereas, after CA addition, the average diameter of fibers increased to a maximum of 0.37 μm for PVDF/CA-1. The diameter dispersion increased, spreading from 0.05 μm–0.40 μm to 0.10 μm–1.0 μm, owing to the increased viscosity of the PVDF/CA polymer solution, which resulted in increased surface tension on the Taylor cone formed at the tip of the needle. With constant electric field forces, a decrease in charged jet stretching was induced, causing a larger diameter and more distribution.

The X-ray diffraction (XRD) spectrum obtained from as-received PVDF powder exhibited characteristic peaks at 18.5°, 20.0°, and 26.6° of the α-phase (black curve in Figure 4a), corresponding to the (020), (110), and (021) crystal planes [34,35], respectively, indicating that α-phase was the main component of the PVDF powder. For pure PVDF fibers, two additional diffraction peaks at 2θ = 20.6° and 2θ = 36.6° appeared, corresponding to the (110) and (200) crystal planes of the β-phase [35], respectively, whereas the characteristic peaks at 2θ = 20.0° and 2θ = 26.6° of the α-phase basically disappeared, and that at 2θ = 18.5° significantly diminished. A comparison of the two results reveals that the electrospinning technique effectively promoted the transition of the α- to β-phase. This conversion may have been due to the polarization and stretching of the high-voltage electric field on the PVDF solution. Moreover, the diffraction peak of the α-phase (2θ = 18.5°) was further weakened, and the intensity of the peak at 20.6° of the β-phase increased for PVDF/CA-1 fiber membranes; when the ratio of PVDF/CA increased, the intensity of the peak decreased. With excess CA, the increased viscosity of the PVDF solution intensified the entanglement between the macromolecular chains in PVDF, which limited the movement of the molecular chains, thereby leading to a decrease in the polarization rate of the solution and a decrease in the β-phase.

As a comparison, the XRD patterns of the PVDF/CA solution cast films in Figure 4b show diffraction peaks at 2θ = 18.5°, 20.0°, and 26.6°, corresponding to the α-phase. This result suggests that the α-phase was the dominant component of the PVDF/CA cast films, while the addition of CA had essentially no effect on the crystal structure of the PVDF cast film. 

FTIR experiments were also conducted to characterize the crystalline phase of the PVDF powder and PVDF/CA EFMs, as shown in Figure 5a. PVDF powder had absorption bands at 975 cm^−1^ and 762 cm^−1^ corresponding to the typical vibration characteristics of the α-phase [36], confirming that the nonpolar α-phase was mainly present in PVDF powder. For pure PVDF fiber, absorption bands were observed at 840 cm^−1^ and 1276 cm^−1^, which were assigned to the polar phases (β- and γ-phases), whereas the α-phase absorption bands substantially weakened, indicating that electrostatic spinning could effectively promote the conversion of the α-phase to β-phase [17]. When loaded with CA, the characteristic absorption peak of CA appeared at 1749 cm^−1^, and the intensity of the β-phase was significantly enhanced. Then, the absorption peaks of the β-phase gradually weakened when the ratio of PVDF/CA was increased (PVDF/CA-2/3). The obtained results prove that an appropriate amount of CA could inhibit the formation of the α-phase and advance the formation of the β-phase.

Figure 5b shows the FTIR spectra of the cast films with CA. It can be seen that the intensity of the absorption peaks of the nonpolar crystals of the α-phase at wave numbers 762 cm^−1^ and 975 cm^−1^ had no obvious change before and after adding CA. The weak absorption peak of the polar β-phase at 840 cm^−1^ was slightly enhanced after adding CA, but the change was not significant. Furthermore, no diffraction peaks from other phases could be observed. Generally, these results confirm that the cast film was still dominated by nonpolar crystals. This is in the agreement with the conclusions drawn from the XRD patterns.

In addition, assuming that the infrared absorption peaks obey the Lambert–Beer law, the content of β-phase in PVDF could be obtained from the following equation [36]: (a)$$F\left(\beta \right)=\frac{A_{\beta }}{\frac{{Κ}_{\beta }}{{Κ}_{\alpha }}A_{\alpha }+A_{\beta }}\times 100\%,$$
where K_α_ (6.1 × 10^4^ cm^2^/mol) and K_β_ (= 7.7 × 10^4^ cm^2^/mol) are the absorption coefficients at 763 cm^−1^ and 840 cm^−1^, and A_α_ and A_β_ are the absorbances of the α-phase at 763 cm^−1^ and the β-phase at 840 cm^−1^, respectively. F(β) represents the relative β fraction. Figure 6 illustrates the values of F(β). The F(β) value of PVDF powder was only 11.26%. The F(β) of pure PVDF fibers prepared via electrospinning significantly increased to 63.65%, while the F(β) values increased to 69.31% in the composite membrane PVDF/CA-1. Upon increasing the content of CA, the F(β) value decreased gradually. 

Positron annihilation lifetime spectroscopy (PALS) is an efficient technique to detect free volumes in the microstructure. The ortho-positronium (o-Ps) lifetime reflects the average free volume hole size in polymers [37,38]. Figure 6b shows the change in measured o-Ps intensity with various samples. The intensity of o-Ps of EFMs was lower than that in cast films, indicating that EFMs contained fewer amorphous regions via cast films. This result may suggest that electrospinning is conducive to crystallinity improvement.

To investigate the piezoelectric properties of the composite fiber membranes, PVDF/CA EFMs were placed in a self-made laboratory pressure device according to the method in Figure 1a. Under a mechanical force of 1 N and an effective area of 4 × 4 cm^2^, the piezoelectric output of PVDF/CA EFM-based PENGs with periodic horizontal compression and release processes was measured using a force simulator. The output voltage and current of the EFM nanogenerator with different PVDF/CA ratios are shown in Figure 7a. It can be seen that the output voltage of the PVDF EFM nanogenerator without CA was only 2.1 V, and the current was 0.9 μA. When the mass ratio of PVDF to CA was 9:1 (PVDF/CA-1), the voltage (4.1 V) and current (1.5 μA) increased by 78.3% and 66.7%, respectively, compared with pure PVDF. However, with increasing CA, a decrease in piezoelectric output was observed. The output electrical signal of the PVDF/CA EFM nanogenerator showed a trend of increasing first and then decreasing with an increase in the proportion of CA, indicating that adding an appropriate amount of CA could improve the piezoelectric performance of PVDF EFMs. The addition of a small amount of CA may have increased the polarization of the spinning solution and, thus, promote polar β-phase generation in PVDF, whereas an excess of CA may have inhibited the orientation along the fiber axis and correspondingly reduced the F(β) value of PVDF/CA [39,40]. Moreover, the piezoelectric output had a positive correlation with the F(β) value in PVDF, which verified that the β-phase in PVDF is the key component of piezoelectricity.

In addition, the effects of some key parameters of PENG, such as mechanical force and membrane area [41,42], were investigated. By adjusting the input power of the motor to change mechanical force, the piezoelectric output could be tested under 1 N and 2 N. As can be seen from Figure 7b,d, the piezoelectric output of the PVDF/CA-1 EFM-based PENG (mass ratio of 9:1) reached the maximum value under either 1 N or 2 N mechanical stresses. Compared with the mechanical force of 1 N, the output voltage and current of the PVDF/CA-1 EFM nanogenerator under 2 N increased by 66.7% and 33.3%, indicating that an increase in the mechanical force could significantly improve the piezoelectric output of the PVDF/CA-1 EFM nanogenerator. This is possibly because, when the piezoelectric material is subjected to mechanical force, bound charges appear on the surface, and the charge density is proportional to mechanical force [43]. Accordingly, a greater mechanical force results in higher piezoelectric properties of the PVDF/CA EFM nanogenerator.

Similarly, we studied the effect of membrane area on the piezoelectric output of PVDF/CA EFM nanogenerators. Figure 7c,e show the trend of output voltage and current of the PVDF/CA EFM devices with membrane areas of 4 × 4 cm^2^ and 8 × 8 cm^2^. When the mass ratio of PVDF/CA was 9:1, the PENG output was best. Compared with the 4 × 4 cm^2^ membrane area, the output voltage and current of EFMs with the 8 × 8 cm^2^ area improved by 77.8% and 53.3%, respectively. The improved piezoelectric performance of PVDF/CA EFM-based PENGs could be attributed to the fact that increasing the area of the membrane promoted the speed of electron transfer, thus inducing a stronger piezoelectric effect [44].

To confirm the superiority of the fabricated PVDF/CA nanogenerator, it is compared in Table 1 with the reported results of other admixtures. In this study, under a periodic stress of 2 N, the output voltage and circuit from the PENG were 7.5 V and 2.1 μA, as depicted in Figure 7b.

The output performance of this PENG was found to be superior to that of other reported PVDF/filler-based devices, as presented in Table 1. This result may be explained by the fact that CA contains a large number of hydroxyl groups (–OH), which have strong interactions such as hydrogen bonds with –CH_2_–CF_2_ in the PVDF molecular chains [33], as shown in Figure 8. Thus, with the polarization-inducing effect of the electrostatic fields, the PVDF chains are induced to transform into the TTTT conformation, indicating a greater conversion of the α-phase to β-phase. According to the results of the FTIR and X-ray diffraction experiments, it can be assumed that, under the optimum concentration, CA can play the role of nucleating agent during the β-phase crystallization process, exhibiting a similar trend to that seen in the literature [33,40]. It is worth noting that the F(β) value of PVDF/CA is not proportional to the CA content. This is possibly because, when the CA content exceeds a certain limit, CA provides a great number of nucleating agents, becoming less oriented along the fiber axis, which leads to a lower content of β-phase.

Nevertheless, for the prepared PVDF/CA cast films, the addition of CA did not result in an obvious transformation. This discrepancy could be attributed to the surface tension of the solution and did not affect the promotion of β-phase formation.

To investigate the driving capability of the PVDF/CA PENG, the AC voltage generated by the nanogenerator was DC-converted to charge a commercial electrolytic capacitor (100 v/4.7 μF) (Figure 9a). Under 1 N and 8 × 8 cm^2^, we charged the capacitors using the PVDF/CA PENG. As shown in Figure 9b, the pure PVDF EFM device could only charge the capacitor to 12.5 V for 8 min. However, the PVDF/CA-1 EFM device could charge the capacitor to 20 V for 8 min with the fastest charging speed, achieving 22 V of charge in 15 min (Figure 9b). This is consistent with the piezoelectric properties of PVDF/CA EFMs with different CA contents. Moreover, the stored circuit, charged by the nanogenerator for 15 min, successfully lit up nine LEDs in series (Figure 9c). This revealed that the PENG has certain application prospects in energy storage and low-power electronic product driving.

The fabricated PENG was also capable of generating piezoelectric output under repetitive compression and release of bending, tapping, palm pressing, and fist beating, as shown in Figure 10a. It can be seen that PVDF/CA nanogenerators showed the corresponding output voltages of 0.2, 2.1, 4.1, and 5.4 V, respectively, revealing it as a potential candidate for harvesting energy from human actions. The piezoelectric voltages of the PENG were 2.6, 9.1, 12.6, and 14.8 V at frequencies of 3, 5, 7, and 9 Hz, respectively, as shown in Figure 10b. The output voltage increased with increasing frequency. However, prior studies noted that output voltage is a frequency-independent parameter [50,51]. The reason for this is that, upon increasing the frequency by adjusting the reciprocating motor, the mechanical force also increases. Furthermore, the PENG was connected to different load resistances ranging from 0.7 to 10 MΩ (Figure 10c). As the load resistance increased to 10 MΩ, the load voltage gradually increased to ~38 V, yielding a maximum output power of 2.26 μW/cm^2^.

## 4. Conclusions

In this work, PVDF/CA EFMs with uniform diameters were synthesized by electrospinning. The composite film output a maximum voltage and short-circuit current of ∼7.5 V and ∼2.1 μA, i.e., 2.5 and two times higher than the output of the pure PVDF fibers, respectively. An increase in the mechanical force and membrane area could also improve the piezoelectric properties of PVDF/CA EFM-based PENGs. The PENG was used as a capacitor (4.7 µF) for charging and LED illumination, in which the capacitor could be charged to 22 V in 15 min, consequently lighting nine LEDs. The great piezoelectric performance of the PVDF/CA EFM nanogenerator highlights its application potential for energy harvesting. This can be attributed to two aspects. Firstly, the PVDF is stretched and polarized during the spinning process due to a high-voltage electric field. Secondly, the formation of hydrogen bond interactions between a large number of hydroxyl groups (–OH) of CA and the fluorine atom of –CH_2_/–CF_2_ in the PVDF molecular chain enhances the polarization of the electrospinning solution. High-voltage polarization and hydrogen bond interactions simultaneously promote the β-phase of PVDF, as well as its piezoelectric properties. 

## Figures and Tables

**Figure 1 materials-15-07026-f001:**
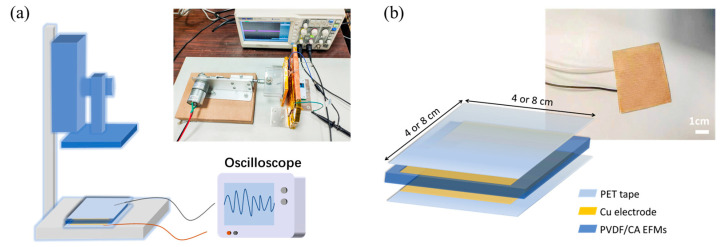
(**a**) The test apparatus; (**b**) schematical illustrations of the structure of the PVDF/CA PENG.

**Figure 2 materials-15-07026-f002:**
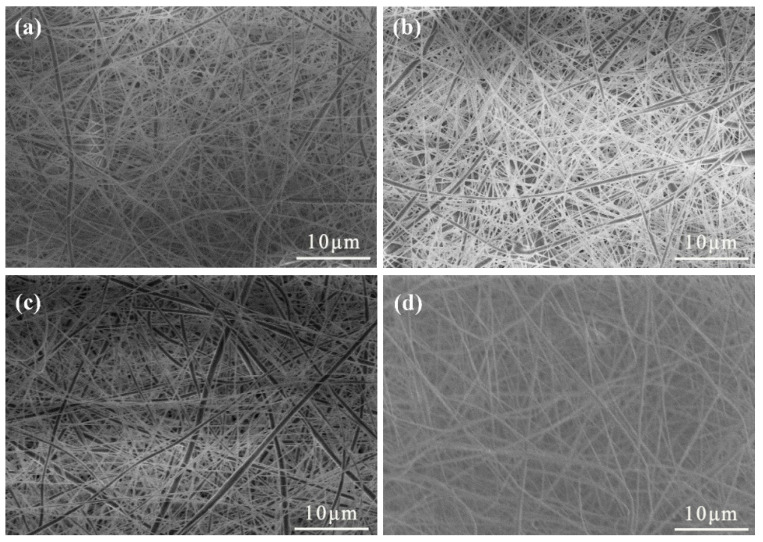
SEM images of PVDF/CA EFMs: (**a**) pure PVDF; (**b**) PVDF/CA-1; (**c**) PVDF/CA-2; (**d**) PVDF/CA-3.

**Figure 3 materials-15-07026-f003:**
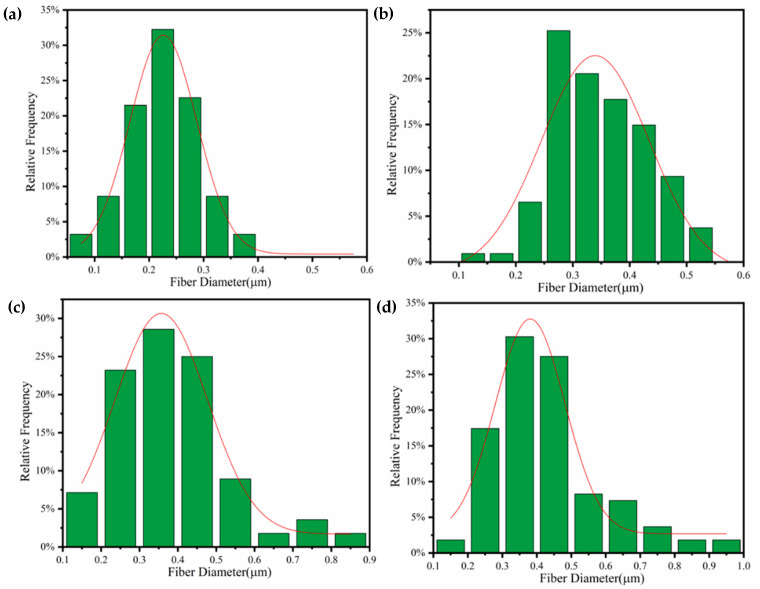
Fiber diameter distribution of PVDF/CA EFMs: (**a**) pure PVDF; (**b**) PVDF/CA-1; (**c**) PVDF/CA-2; (**d**) PVDF/CA-3.

**Figure 4 materials-15-07026-f004:**
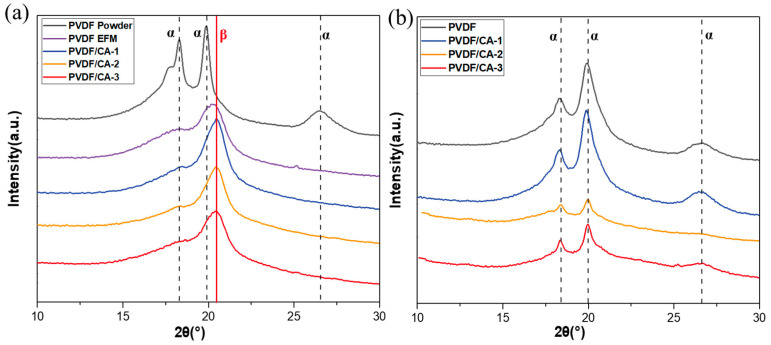
XRD analysis of (**a**) PVDF/CA EFMs and PVDF powder, and (**b**) PVDF/CA cast film.

**Figure 5 materials-15-07026-f005:**
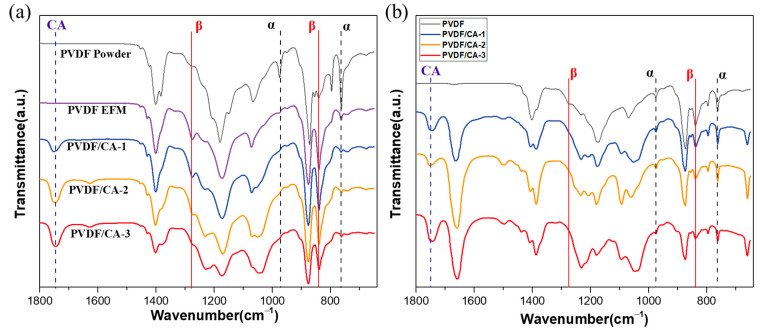
FTIR spectra of (**a**) PVDF powder and PVDF/CA EFMs, and (**b**) PVDF/CA cast film.

**Figure 6 materials-15-07026-f006:**
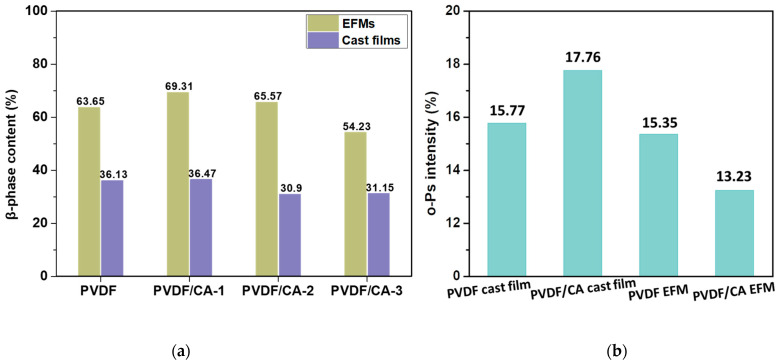
(**a**) The F(β) value of PVDF/CA samples; (**b**) ortho-positronium intensity.

**Figure 7 materials-15-07026-f007:**
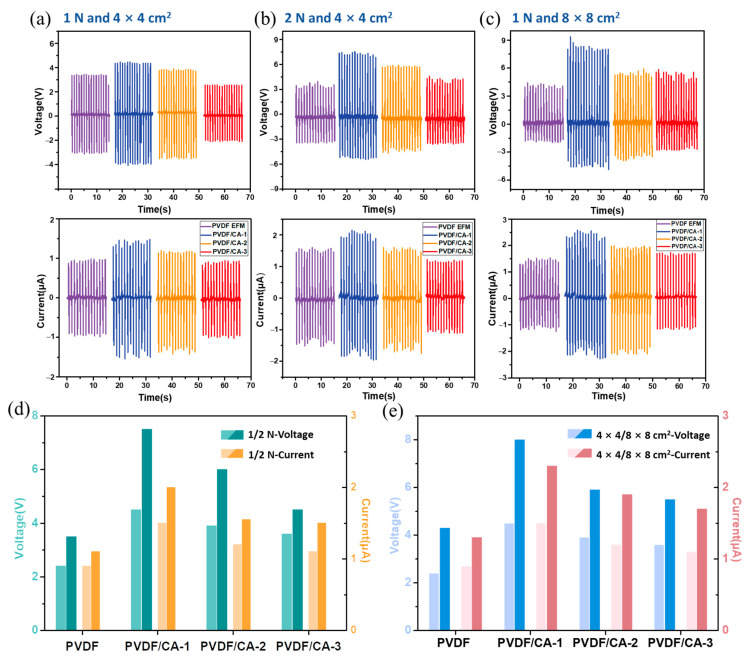
Output voltages and currents of PVDF/CA EFM-based PENGs with a membrane area of 4 × 4 cm^2^ (**a**) under 1 N, and (**b**) under 2 N; (**c**) output voltages and currents of PVDF/CA EFM-based PENGs with a membrane area of 8 × 8 cm^2^ under 1 N; trend diagrams of maximum output voltages and currents of PVDF/CA EFM-based PENGs (**d**) under 1 N and 2 N, and (**e**) and with membrane areas of 4 × 4 cm^2^ and 8 × 8 cm^2^.

**Figure 8 materials-15-07026-f008:**
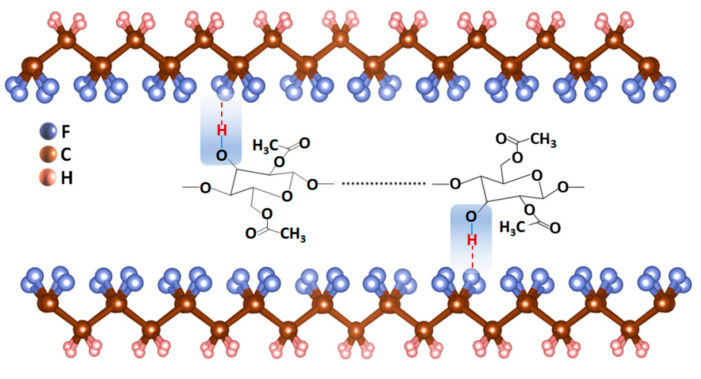
The chemical interactions between the CA and PVDF molecular chains.

**Figure 9 materials-15-07026-f009:**
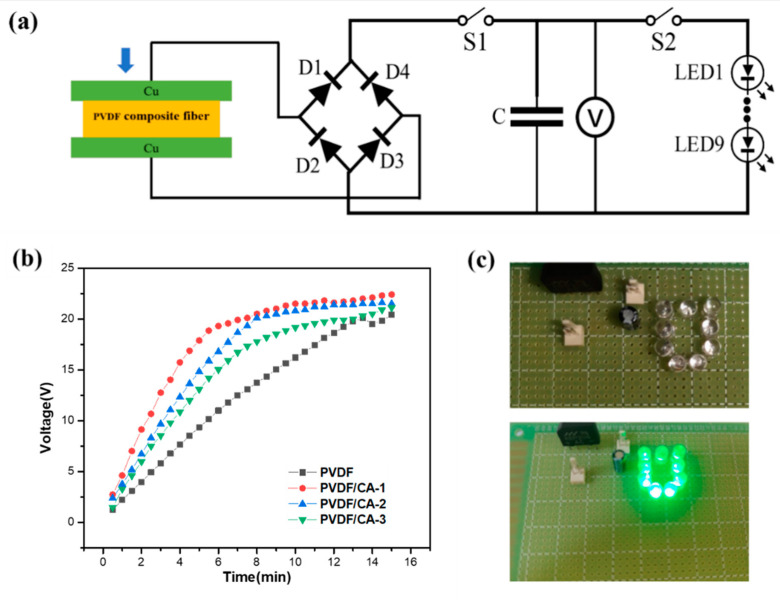
Piezoelectric response of PVDF/CA EFMs: (**a**) the circuit diagram of the commercial capacitor charged with PVDF/CA EFMs-based PENGs; (**b**) voltage–time curves of capacitor charging; (**c**) LED lighting driven by capacitive discharge with PVDF/CA EFM-based PENGs.

**Figure 10 materials-15-07026-f010:**
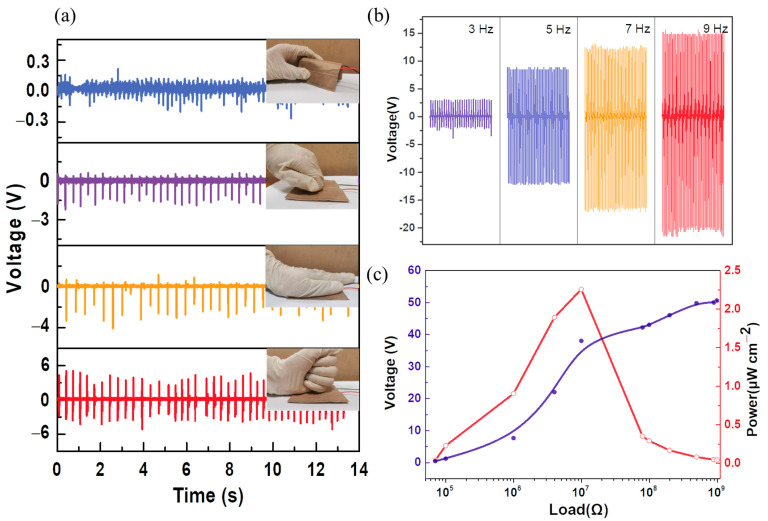
(**a**) Output voltage generated by the PENG under bending, tapping, palm pressing, and fist beating; (**b**) output voltage with frequency; (**c**) output voltage and power across varying resistances from the PENG upon exposure to constant force and frequency at 1 N and 1 Hz.

**Table 1 materials-15-07026-t001:** Comparison of the performance of PVDF-based piezoelectric nanogenerators for different admixtures.

Materials	Synthesis	Size (cm^2^)	Pressure (kPa)	Voltage (V)	Current (μA)	Ref.
ZnO/PVDF	Spin coating	–	–	0.41	0.029	[45]
PZT/PVDF	Hot pressing	0.503	85.59	2.51	0.07843	[46]
CNC/PVDF	Electrospinning	4	–	6.3	–	[40]
RGO/PVDF	Solution casting	10	10-12	1.915	–	[47]
Graphene/PVDF	Electrospinning	4	200	7.9	4.5	[48]
RGO/NaNbO_3_/PVDF	Solution casting	4	15	2.16	0.383	[49]
BTO@HBP@PMMA/PVDF	Electrospinning	9	44.44	3.4	0.32	[8]
PVDF/CA	Electrospinning	16	1.25	7.5	2.1	This work

## Data Availability

Raw data are available upon request.

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
