# Peer review of "Piezoelectric Nanogenerator Based on Electrospinning PVDF/Cellulose Acetate Composite Membranes for Energy Harvesting"

_materials, 2022, doi:10.3390/ma15197026_

Round 1

Reviewer 2 Report

In my opinion, the manuscript is suitable for publication in Materials journal but Authors must complete a major revision. Manuscript should be revised according to following comments:

1. Chapter “Introduction” must be improved:
a) The authors must present the basis on which they chose Cellulose Acetate for fiber production. Literature sources or preliminary research should be presented.

2. Chapter “Materials and Methods” must be improved:
a) Figure 1: the drawing is too general and does not add scientific information. The drawing should be completed:
- indication of the most important dimensions,
- readability should be greatly improved,
- this drawing does not provide scientific information. They suggest that you refer to the various parts shown in figures a and b.
b) apparatus parameters should be given:
- sensor names and their parameters: resolution, measuring range,
- equipment used for data acquisition and its parameters,
c) a photo of the laboratory stand where the tests were performed should be added.

3. Chapter “Results and Discussion” must be improved:
a) the time course of the external load applied to the PVDF / CA for the laboratory stand shown in Figure 8a should be added,
b) The authors present in Figure 8b the results for one type of external load. Comparative tests should be carried out for different (both in terms of amplitude and frequency) external load waveforms,
c) the authors compare the properties of the developed PVDF / CA material with PVDF without admixtures. The authors should compare developed material to PVDF with other than CA admixtures in the discussion,
d) the authors should designate the electromechanical coupling factor "d" for the developed material.

Reviewer 3 Report

Here, the electrospinning technique was taken to fabricate the synthesized nanofiber membranes of PVDF/cellulose acetate (CA) composite. The obtained PVDF/CA electrospinning fiber membranes (EFMs) were used to prepare a flexible nanogenerator.

The subject is relevant, but the way it is approached does not make it publishable in its current form.

The authors need to carry out further analysis to demonstrate the improvement of the piezoelectric response after the addition of CA on PVDF.

PFM studies should be carried out, also it should be clarified if the improvement of piezo properties is not attributed to electrostatic effects.

Nanogenerator performance should be tested at different frequencies.

Attention at:

-  to indicate figs 4 and 5,  a) and b)  such that the reader can know which one is

- refer in the text to all figures appearing in the manuscript indicating them accordingly

Wang, X., Yang, B., Liu, J. et al. A flexible triboelectric-piezoelectric hybrid nanogenerator based on P(VDF-TrFE) nanofibers and PDMS/MWCNT for wearable devices. Sci Rep 6, 36409 (2016). https://doi.org/10.1038/srep36409

Reviewer 4 Report

In this work, the authors prepared the PVDF/cellulose acetate composites by electrospinning. Cellulose acetate was added to PVDF ferroelectric polymer to enhance the β-phase content and thus, the piezoelectricity. Morphology and physicochemical properties of the composite membranes with different cellulose acetate contents were studied. They adopted the PVDF/cellulose acetate composites to fabricate PENGs.

The research study is consistent and reasonably laid out. The report is well presented and clearly written. While cellulose-based materials are non-toxic, cellulose can be very reactive with water molecules. Therefore, the electromechanical properties or functionality of these composites may be degraded with time, which is my main concern. Can the authors make some comments?

Round 2

Reviewer 1 Report

The overall structure of the paper has been efficiently revised and I believe it's now in accordance with the standard of the journal. Good work!

Reviewer 2 Report

I accept in present form.